# Impact of Almond Variety on “Amaretti” Cookies as Assessed through Image Features Modeling, Physical Chemical Measures and Sensory Analyses

**DOI:** 10.3390/foods9091272

**Published:** 2020-09-10

**Authors:** Marco Campus, Manuela Sanna, Giandomenico Scanu, Riccardo Di Salvo, Luciano De Pau, Daniela Satta, Antonio Demarcus, Tonina Roggio

**Affiliations:** 1Porto Conte Ricerche S.r.l., 07041 Tramariglio (SS), Italy; sanna@portocontericerche.it (M.S.); roggio@portocontericerche.it (T.R.); 2AGRIS, Agricultural Research Agency of Sardinia, 07100 Bonassai (SS), Italy; gdscanu@agrisricerca.it (G.S.); rdisalvo@agrisricerca.it (R.D.S.); ldepau@agrisricerca.it (L.D.P.); dsatta@agrisricerca.it (D.S.); 3Independent Researcher, Environmental Engineer, 07016 Pattada (SS), Italy; antonio.demarcus@gmail.com

**Keywords:** almond, sweet cookies, amaretti, physical chemical changes, image analysis, texture, sensory profile, consumer test

## Abstract

In the present study, the influence of almond variety on color, chemical, physical and sensory characteristics of “amaretti” cookies during the shelf life, was assessed. Four varieties were chosen for the study, two of which were local (Cossu, Arrubia) and two widely cultivated (Tuono, Texas). Almonds have been characterized in the content of proteins, crude fat, amygdalin and fatty acids profile. The evolution of the characteristics during the shelf life hasbeen measured through image data modeling, texture, physical chemical and sensory analyses. Data were then treated with a multivariate approach performing a PCA. Image analysis and fitting on log normal and powerlaw functions highlighted the influence of the variety on the total area affected by surface breakages, and on the distribution of the cracking surfaces dimension classes. Texture parameters (crust hardness, thickness and work of deformation) were negatively correlated to moisture content. Sensory profile confirmed the differences in tactile features measured through instrumental texture, while slight to no differences were found in odor profile. Consumer test showed an higher acceptability for Arrubia, Texas and Tuono samples throughout the shelf life, while Cossu samples were less accepted. Overall, the choice of almond variety influences product features and liking of almond products, therefore it represents an important phase to direct the choice of both farmers and confectionery manufacturers.

## 1. Introduction

“Amaretti” cookies are a renowned Italian pastry, appreciated for their bitter-sweet taste and almond flavor. The amaretti recipe is quite simple, since they are obtained by grinding and mixing together sweet and bitter almonds (or alternatively using almond extract), sucrose and egg white. The cookies are characterized by a delicate almond flavor, a crispy crust, with low water activity (a_w_), and a softer inner part, which retains most of the moisture [1]. Therefore, according to the definition by Labuza and Hyman [2], amaretti can be considered a macromolecular multi-domain system. Regions at different water activities in multi-domain foods cause the whole system to be in a non-equilibrium state. This results in moisture migration from the higher a_w_ to the lower a_w_ region. Shelf life limiting factors of amaretti cookies are the progressive firming, due mostly to moisture loss and sucrose recrystallization, during storage [3]. As a consequence, texture may move from rubbery to grainy. Sucrose recrystallization is related to moisture migration between components, and is accompanied by an increase in a_w_ values [4], with the potential exposure to microbial spoilage (mold growth), as a_w_ approaches 0.8. Packaging solutions [3,5] and addition of fortifiers [6] have been studied to extend the shelf life of almond pastries. No contributions have been reported on the impact of different almond varieties on the shelf life and acceptability of this type of product. Almonds are characterized by a high lipid and protein contents. Lipids may act as moisture barriers to moisture migration in a multi-domain system, largely depending on their composition, the pore size in the matrix of the food domain, crystals or lipid interferences in the system. Fat acts as a lubricant and contributes to the plasticity of the cookie dough, imparts desirable eating qualities and contributes to texture and flavor of the product. Proteins can interact with water, from loose interactions to structural water entrapped in proteins structures, which could be unavailable for chemical reactions [7]. Starting from the premise that differences between varieties could affect the final product structure and composition and thus the shelf life limiting phenomena, the aim of the present paper was to assess the differences in amaretti cookies prepared using different almond varieties. Moreover, surface cracks, such as those found in this type of product, can play a role in chemical physical changes and stability. In this regard, we have developed a workflow for image analysis, based on the use of free software for image pre-processing, and a MATLAB^®^ (MathWorks, Inc., Natick, MA, USA) code for automation of operations. The procedure allows to have a description of the frequencies of the dimensional classes of the surfaces affected by breakages. We will illustrate the math together with the results, while the MATLAB^®^ code will be made available on Github website (https://github.com/AntonioDEM/powerlawlog_project). Instrumental texture, chemical-physical measures and sensory characteristics were assessed to monitor the evolution of shelf life parameters during shelf life. Variables were subjected to PCA to identify patterns in data based on the correlation between features.

## 2. Materials and Methods

### 2.1. Almond Origin and Processing

Almonds were collected from the almond germplasm field collection of the Agricultural Research Agency of Sardinia (AGRIS), located in Sardinia (Italy). The soil is sandy-clay (42% sand), pH 7.4, the average annual precipitations (5-year average) were 475.4 mm The plants of the studied varieties, grafted onto GF677 rootstock, are planted at a 6 m × 6 m distance, trained to a goblet shape at 80 cm trunk height. Winter and green pruning are provided annually, and irrigation is supplied using two self-compensating drip rows buried at 30 cm depth. In each dripline emitters (4 l/h) are spaced 40 cm, and about 3.500 m^3^/ha of irrigation water are provided to the plants each year. Harvest took place from the end of August until the first days of September 2018. The fruits were manually collected from the ground at full ripening. Hulling and shelling operations were conducted mechanically, then the almonds were peeled by dipping the fruits in boiling water until skin removal was achievable by hand. After that, moisture was rebalanced for all varieties to reach values below 6% by air drying in a desiccator (36 °C). Four varieties, of which two (Texas, TX; Tuono, TU) are widely diffused commercial varieties and two (Arrubia, AR; Cossu, CO) are local varieties were evaluated for their attitude to industrial transformation. Peeled almond kernels have been used for both making cookies and chemical analyses.

### 2.2. Almond Composition

#### 2.2.1. Moisture Content

For each variety, 35 whole peeled almonds (about 50 g) were sampled. Almonds were ground and sifted with stainless steel sieves 18 (1 mm pore size). Crushing lasted less than 30 s, in order to avoid sample heating and agglomeration. Five g of wet almond flour were weighed with an analytical balance (E42S-B, Gibertini, Novate Milanese (MI), Italy) to determine. Then, almond flour was desiccated at 105 °C until constant weight. After cooling the samples in a vacuum dryer for 15 min, dry weight was recorded and moisture content calculated.

#### 2.2.2. Ether Extract

For crude fat content determination, a Buchi B-811 extractor (Buchi, Essen, Germany) was used, performing a Soxhlet type automatic extraction. About 4 g of dried flour were introduced into the extraction thimble, closed at the end with a cotton pad. 120 mL of petroleum ether 40–60 °C RPE (Carlo Erba Reagents, Cornaredo (MI), Italy) were poured in the system and the standard Soxhlet program was set up with 13 extraction cycles for each station and 5 min of rinsing (without drying). At the end of the extraction, the flour was placed in oven at 105 °C for one night, in order to remove the solvent. Then the samples were weighed and the fat content calculated.

#### 2.2.3. Fatty Acid Profile

Fatty acids methyl esters (FAME) were prepared by dissolving 0.5 g of oil in 6 mL of *n*-hexane, then adding 0.25 mL of KOH 2N in methanol. After 10 s of vigorous shaking, samples were centrifuged at 3000 rpm for 10 min and the supernatant collected. FAME analysis was done with an Agilent 7890A Gas Chromatograph (Agilent, Palo Alto, CA, USA), equipped with a flame ionization detector (FID). Separation was carried out with a, SP-2380 Capillary GC Column (Supelco, Bellefonte, PA, USA; 60 m × 0.25 mm i.d.; 0.2 µm film thickness) using Helium as the carrier gas, at a flow rate of 1.2 mL/min. The GC oven temperature program began with the oven held at 185 °C for 17 min, then increased to 220 °C at 4 °C/min, maintained at 220 °C for 8 min, then to 230 °C at 2 °C/min, for 6 min. The total run time was 45 min. Detector temperature was set at 300 °C, H2 flow at 30 mL/min, air flow at 400 mL/min, make up gas (N2) flow at 25 mL/min. Sample Injection (1 μL) was made in Split mode (50:1) at 60 mL/min. FAME were identified comparing retention times with those of authentic standards (Sigma Aldrich, Saint Louis, MO, USA) and expressed as area units (%) in respect to the total total ion current (TIC) area.

#### 2.2.4. Protein Content

Proteins content was assessed on defatted samples according to the bicinchoninic acid assay (BCA) method using a total protein colorimetric assay Kit (Bio-Rad Protein Assay Kit, Bio-Rad, Hercules, CA, USA).

#### 2.2.5. Amygdalin Content

Amygdalin extraction was performed adding 5 mL of MeOH (Ultra Gradient HPLC Grade JT Baker, Kansas city, NJ, USA) to 0.4 g of ground almonds in 20 mL flasks. The mixture was stirred in water bath at 30 °C for 16 h. Subsequently, the mixture was centrifuged (with a Heraeus Megafuge 40R centrifuge, Thermo Scientific, Waltham, MA, USA) at 4500 rpm for 15 min; the supernatant obtained was then filtered with nylon filters (OlimPeak0.45 µm, 25 mm Ø, Teknokroma, Barcelona, Spain) [8]. Finally, 1000 µL of sample were taken and transferred to vials for HPLC analysis of TU, CO, and TX varieties almond flour, while for AR (which had supposedly a higher amygdalin content) the sample was diluted in MeOH with a ratio of 1:2 before HPLC analysis. Analytical HPLC grade standard of amygdalin was purchased from Sigma Aldrich and used to construct a 7-point calibration curve. The initial concentration of 33.85 ppm was chosen after preliminary analysis. For each point of the line, three standard dilutions were prepared, each injected in triplicate. An Alliance HPLC instrument (Waters, Milford, MA, USA) equipped with an e2695 XC separation module, PDA 2998 detector and Gemini 3U C18 110A column (150 × 4.60 mm 3 µm, Phenomenex, Torrance, CA, USA) was used. Starting from an original mixture of CH_3_CN 20:80 [8,9] with a column temperature of 42 °C, the following parameters were set: injection volume: 20 µL, run time: 5 min for standards and 6 min for samples, wavelength (λ): 218 nm, mobile phase: isocratic with unique mixture 21% CH_3_CN (acetonitrile RS, Carlo Erba Reagents for HPLC-isocratic grade) and 79% ultrapure H_2_O, mobile phase flow: 1.00 mL/min. At sample setting end, the column system was flushed with MeOH, gradually lowering the flow (from 1 to 0 mL/min).

### 2.3. Amaretti Cookie Quality Characteristics

Two batches of “amaretti” cookies were produced in a local bakery. The following recipe was adopted, expressed as g per 1000 g of dough: 397 g of sucrose, 341 g of sweet almonds, 190 g of egg white (pasteurized), 68 g of bitter almonds, 4 g of citrus aroma. Sweet and bitter almonds together with sucrose (commercially available as granulated sugar and derived from sugar beet, with a particle size under 700 mm) were grinded in a bakery grinding machine. Egg white was then gradually added to the mixture. The ingredients were mixed for 15 min. After formation amaretti were cooked at 160 °C for 40 min in a static oven (Real Forni Srl, Verona, Italy). Baking was followed by cooling at room temperature, then cookies (disc of 35 mm Ø, 30 mm high and weighing 25 gr) were packed in card trays with topping of shrinkable polyolefin film (19 μm). The packaged samples were stored at 23 °C and 65% RH, in the dark, until analyses.

#### 2.3.1. Image Analysis of Crackings Surfaces

Images of “amaretti” cookies were captured using a VersaDoc 4000MP system (Bio-Rad), obtaining images in 24-bit color, at a resolution of 400 dots per inch, i.e., 1 pixel = (60)2 μm^2^. For each variety, 30 cookies were analyzed. Image processing was performed using ImageJ 1.51K (Rasband, W.S., ImageJ, National Institutes of Health, Bethesda, MD, USA). The images were pre-treated by conversion into grayscale images, then converted into 8-bit black and white binary images. To optimize the image analysis, Otsu thresholding method was applied. Due to convexity of samples, a planar portion of the surface was selected, capturing a 5 × 5 cm circle from each cookie, taken from the center of the image, as shown in Figure 1. 

The binary images were analyzed for particles number and sizes, using ImageJ particle analysis routine. The output of the analysis was the number of “cracking areas” (CKA) and their plain size (area, in mm^2^). Data for the areas and their frequency were used to compute the cumulative distribution function (CDF), the complementary cumulative distribution function (CCDF), and probability distribution function (PDF), fitting the entire CCDF distribution to a log-normal PDF, and then using the power-law PDF for the tail of the distribution. All the procedure was carried out using MATLAB^®^ (MathWorks, Inc., Natick, MA, USA). For the convenience of the reader, the math used will be detailed along with the results hereinafter in the article.

#### 2.3.2. Color Measurements

The color was measured on 30 cookies per variety, taking three measurements from each cookie, one from the center and two from the external parts, using a CM-700d spectrophotometer (Konica Minolta, Osaka, Japan), using Standard Illuminant D65/10°. Prior to measurements, the Instrument was calibrated against the white tile. CIE L*a*b* color space coordinates, lightness (L*), color in the red/green field (a*) and color in the blue/yellow field (b*), were computed. The differences in lightness (ΔL′), Chroma (ΔC′), Hue (ΔH) and Hue angle (Δh) were calculated and used to elaborate the ΔE00, the Euclidean distance between colors, as recommended by CIE (2001), using the following formula:(1)$$\Delta E_{00}=\sqrt{{\left(\frac{\Delta L{}^{\prime }}{k_{L}S_{L}}\right)}^{2}+{\left(\frac{\Delta C{}^{\prime }}{k_{C}S_{C}}\right)}^{2}+{\left(\frac{\Delta H{}^{\prime }}{k_{H}S_{H}}\right)}^{2}+R_{T}\frac{\Delta C{}^{\prime }}{k_{C}S_{C}}\frac{\Delta H{}^{\prime }}{k_{H}S_{H}}}$$

For the detailed explanation of computed parameters, see [10]

The corresponding ΔE76 values were used to estimate the range of perceived difference between samples of close chroma [11]):0 < Δ76 < 1-the difference is unnoticeable1 < Δ76 < 2-the difference is only noticed by an experienced observer2 < Δ76 < 3.5-the difference is also noticed by an unexperienced observer3.5 < Δ76 < 5-the difference is clearly noticeable5 < Δ76-gives the impression that these are two different color

#### 2.3.3. Water Activity (a_w_) and Moisture Determination

a_w_ and moisture were determined in triplicate on six ground amaretti cookies from the same batch, at 1, 7, 15, 30, 60 days of storage. The analyses were performed on whole cookies, due to the difficulty in separating the inner and outer part. Moisture content of amaretti cookies was determined putting the grinded samples (1.5 gr.) in a ventilated oven at 105 °C until constant weight. a_w_ determinations were performed using an AQUALAB instrument (Series 3, Decagon, Pullman, WA, USA), calibrated in the range 0.1–0.95 with solutions of LiCl, NaCl and KCl of known activity [12].

#### 2.3.4. Texture Measures

Texture evolution over time was determined with a texture analyzer (TA.XT Plus, Stable Microsystems, Surrey, UK) equipped with a 25 kg load cell and Texture Expert Exceed software, version 2.64a. Analyses were performed at 1, 7, 15, 30, 60 days of storage on six amaretti cookies per batch and time. A puncture test was carried out with a 5 mm diameter cylinder probe (mod. P/5). Samples were placed in the confectionery holder, supplied with a 6mm diameter top and bottom hole (punch test). This holder allows complete penetration of the probe into the sample avoiding sample displacement at the same time. Samples were placed centrally on the holder and secured on the heavy duty platform before the test. The sample was punctured right through. The following test parameters were set: pretest, test and post-test speeds were, 2, 1 and 5 mm/s, respectively. During the test the probe was lowered 20 mm, and after the test it returned to its start position. Three main parameters were computed: Hardness of the upper crust, as the maximum force (N) reached during puncturing; the work of deformation (WOD), as the area under the curve (N mm) between the reaching of the maximum force mm and the complete probe penetration; the thickness of the upper crust, as the distance (mm) between the starting point of the test and the onset of maximum force.

#### 2.3.5. Sensory Analyses

##### Panel Training

The panel was trained in accordance with the ISO standards [13,14]. The panel was made up of nine expert judges, five females and four males, aged between 35 and 50, with previous experience in the sensory profiling of fruits.

##### Sensory Profile

The sensory descriptive technique was applied to the amaretti cookies [15]. Amaretti produced with the four almonds varieties (TX, TU, AR and CO) were analyzed at 1, 7, 30 and 60 days of storage. The samples, previously acclimatized in a thermostatically controlled oven (20 °C), placed in containers of odorless material, marked with a three-digit random number [16] were presented to the judges in a randomized and balanced order [17], in a tray containing also a cracker and a glass of water, as palate cleansers between samples. Tests were performed in tasting booths [18]. Judges evaluated the intensity of 18 amaretti’s attributes on a 10 cm unstructured scale, from 0 (low perception) to 10 (high perception), for each attribute identified. Three attributes belong to the visual and tactile characteristics (color, roughness and tactile-hardness), 4 to the olfactory (amaretto, citrus fruits, sweets, odor, and caramel), 2 to the taste (salty and bitter), 4 to the aromatic (amaretto, citrus, bitter and off-flavor almond) and 5 to the texture (hardness, friability, humidity, adhesiveness and solubility). Two experimental replicates were performed for each control point. To measure the analytical reliability of the panel’s response, two samples of amaretti cookies were replicated for each shelf-life sampling points. The acquisition of sensory data was carried out using a specific computerized application [19].

##### Consumer Testing

An acceptability test was performed [15]. The acceptability test was carried out by 60 consumers, 30 women and 30 men, most of them recruited on the basis of interest and willingness. They were regular consumers of the product, aged between 32 and 60 years, not trained in the sensory analysis. Consumers were asked to give a score to the following attributes: flavor, taste, texture, appearance and overall acceptability. A nine-point structured hedonic scale ranging from 1 (extremely disliked it) to 9 (extremely liked it) was used, and sample was considered acceptable when scoring above 5 (neither like nor dislike).

### 2.4. Statistical Analyses

Chemical, physical chemical, texture, and sensory data were subjected to analysis of variance (ANOVA) and Tukey test (*n* = 5, *p* ≤ 0.05) as the post hoc test, using the Statgraphics Centurion software package (version 16.1.11, StatPoint Technologies Inc., Warrenton, VA, USA). Image analysis data were processed using MATLAB^®^ routines, using Kolmogorov-Smirnov statistics to evaluate the goodness of fit of CKA on log-normal and power-law distributions. The panel’s judgments, in terms of reproducibility and discriminant ability, was monitored through Three-way ANOVA model (judge, sample and replicates effect) with interaction. The hedonic scores collected from the consumer test were examined using ANOVA and Tukey tests (*p* ≤ 0.05), with consumers (random effect) and products as the main effects. The differences between samples were analyzed by principal component analysis (PCA) of the correlation matrix of selected variables. The component loadings were calculated as simple correlations (using Pearson’s r) between the components (i.e., the component scores) and the original variables.

## 3. Results

### 3.1. Almond Composition

Proximate composition of almond samples are reported in Table 1. Some differences in moisture content were found, with TX samples showing the higher values. Protein content was the same for all samples, except TX, which had slight lower values. Major differences in crude fat content were found between AR and TX, where the minimum and maximum values were found respectively.

Fatty acid profiles are reported in Table 2. The fatty acid composition of the oil may differ depending on the variety. The monounsaturated fatty acids have great importance because of their nutritional implication and effect on oxidative stability of oils. Oleic acid (C18:1) is the main monounsaturated fatty acid and is present in higher concentrations (70.08–75.79%). Palmitic acid (C16:0) is the most abundant saturated fatty acid in almond oil. Concerning linoleic acid (C18:2), which is much more susceptible to oxidation than monounsaturated fatty acids, was observed to have the highest percentage in AR (20.58%), whereas the lowest one was found in TX (15.53%). Regarding the other fatty acids—palmitoleic (C16:1) and stearic (C18:0)—although present in small amounts, their content varied between oil samples. No differences were found in the content of Linolenic acid (C18:3). The ratio between monounsaturated fatty acids (MUFA), and polyunsaturated fatty acids (PUFA) can be useful for cultivars characterization and oxidative stability. Its values ranged from 3.43 (AR) to 4.93(TX). Amygdalin, the bitter glucoside of almonds, was significantly higher in AR, while in TX samples the concentration was below the detection limit.

### 3.2. Image Analysis of Cracking Surfaces

Spatial segmentation for feature extraction is able to provide an estimation of cracking fineness, and also to accurately measure various structural parameters [20]. Otsu algorithm has been applied for the thresholding of acquired images, since it generates good and consistent binary images in terms of thresholding performance measures and features computed [21]. Particle analyses performed over binary images showed that the size (area) of the CKA covered a range from 0.018 to 450 mm^2^. The total area (in mm^2^) interested by cracking computed on 30 cookies were 14,562 (CO), 14,463 (TX) 13,927 (TU) 13,376 (AR), respectively. Large CKA showed a typical profile of a power-law distribution, or more in general, a fat-tail distribution. Data for the areas and the numbers of areas were used directly to compute the complementary cumulative distribution function (CCDF) [22]. In the formulae reported below, the CCDF is defined as (1−CDF), where the CDF (cumulative distribution function) is the integral of the probability distribution function (PDF). With the PDF denoted by *p(x)*, the CDF by *P(x)*, and the CCDF by *G(x)*, where x is the area in mm^2^, the following formulae are given:(2)G(x)=1−P(x)
(3)$$P(x)=p(X\leq x)=\int_{-\infty }^{x}p(x{}^{\prime })dx{}^{\prime }$$
(4)$$G(x)=p(X\geq x)=\int_{x}^{\infty }p(x{}^{\prime })dx{}^{\prime }$$

The best fitting procedure was applied to analyze CKA sizes distribution. An automatic Matlab^®^ application for fitting “log-normal” and ‘power-law’ distribution to empirical data, following the goodnes-of-fit based approach has been used, and is avaiable for download (Power-law-log Project, https://github.com/AntonioDEM/powerlawlog_project). The entire *G(x)* was plotted on log scale and fitted to a Log-normal and Power law distributions:

The log-normal distribution was:(5)$$p(x)=\frac{1}{\sqrt{2\pi \sigma x}}e^{-{\left(\frac{\ln(x)-\mu }{2\sigma }\right)}^{2}}$$

The power-law distribution was:(6)$$p(x)≅cx^{-\alpha }$$

Two parameters are needed to specify the log-normal distribution properties. Usually, the mean μ and the standard deviation (or the variance σ^2^) of log(x) are used. However, there are advantages to using “back-transformed” values, i.e., the values are in terms of x, the measured data [23]:μ* = e^μ^(7)
σ* = e^σ^(8)

μ* = e^μ^, is the median of the log-normal distribution, and also the geometric mean of the distribution in terms of original data. The parameter σ*, the geometric standard deviation, determines the shape of the distribution. Since both μ* and σ* are in the units of the original measurement, these are more easy to interpret and can also describe the log-normal distribution: 68.3% of the distribution is contained between (μ*/σ*)and (μ*·xσ*), 95.5% is contained between (μ*/(σ*)2) and (μ*·(σ*)2) and 99.7% is contained between (μ*/(σ*)3) and (μ*·(σ*)3). During the study, we investigated the presence of a power law component on the “tail”, for *x* greater than a minimal value “xmin”, given that CCDF plotted on logarithmic scale on both axes has a linear decrease component on the “tail” (Figure 2). To estimate the distance between the two distributions, the empirical and theoretical power law, we used the Kolmogorov–Smirnov (KS) statistic D [24]:(9)$$D=\max_{x\geq x\min}|S(x)-\bar{F(x)}|$$
where *S*(*x*) is the empirical CCDF, while *F*(*x*) is the theoretical CCDF of the power law model which best fits the empirical data for *x* ≥ *x*min. The *x*min value estimated is chosen in a way that the estimated power law model gets a best fit of the empirical probability distribution for *x* ≥ *x*min [25,26]. For each possible choice of xmin, the MATLAB^®^ function (plfit.m function, available at http://www.santafe.edu/~aaronc/powerlaws) estimated “alpha” via the method of maximum likelihood, and calculate the Kolmogorov-Smirnov goodness-of-fit statistic, D. We then select as our estimate of xmin, the value that gives the minimum value D over all values of xmin. The D value was tested against a tabulated maximum value, for a given “n” as the sample size [27]. The results of the best fitting procedure over CCDF are depicted in Figure 2, were x (CKA in mm^2^) and CCDF were plotted on a log-log scale. Coefficient arising from the procedure are reported in Table 3. The computed *p* values, obtained from the KS test for the log-normal fits (which resulted < 0.05 for all curves), and D value for the power law, confirmed the log normal distribution of the overall data (small to medium areas) and a power-law distribution in the tails (large areas). ANOVA analysis of coefficients showed no differences in the power-law exponent “α”, indicating that there were no differences in large CKA size distribution between samples. Differences were found in the log-normal coefficients μ*, showing that small to medium CKA distribution were different between samples (Table 3).

### 3.3. Colormeasures

Color analyses results are reported in Table 4. From a paired comparison of color parameters computed, color difference ΔE76 is noticeable at human sight [11], for TX-CO, TX-AR, TU-AR, CO-AR pairs. AR samples are those who showed the major differences with respect to the other samples, and the overall difference is more marked between AR and CO samples. AR is characterized by the lower values of L*, and the resulting ΔL′ is higher for TX-AR, TU-AR and CO-AR pairs; Difference in chroma (ΔC′), Hue (ΔH) and Hue angle (Δh) resulted higher for TX-CO, TU-CO and CO-AR pairs. The color of cookies is related mostly to Maillard reactions and caramelization, which occur during cooking.

### 3.4. Water Activity (a_w_) and Moisture Determination

a_w_ differences between samples were found at the intermediate time of analyses, while at the beginning and at the end of the experiments the samples do not present significant differences (Figure 3). 

All samples showed a noticeable increase in a_w_ values from 1 to 15 days. After that the a_w_ remained almost constant, decreasing in the final analysis point (60 days). a_w_ changes are related to sucrose recrystallization phenomena occurring in intermediate moisture foods containing sucrose [2,4]. Just after cooking, sucrose is in the amorphous metastable state. Amorphous sugars are highly hygroscopic and adsorbed water from the surroundings. This led amorphous sucrose to behave as a supersaturated solution, which favors the sucrose recrystallization, since increasing amounts of moisture decrease the glass transition temperature below room temperature. Sucrose recrystallization frees bound water, with an increase in a_w_ values. This free water is gradually lost due to humidity gradients between headspace, crust and inner part of cookies. The presence of other molecules (proteins) that may share water with sucrose, constitutes an additional factor of instability of such parameters.

Moisture content (Figure 4) remained almost constant to 7 days, decreasing significantly after that point and reaching its minimum at 60 days. The major differences between samples were found at the intermediate time of analysis, in which TX and CO showed the lower values among samples test.

### 3.5. Texture Measures

Amaretti cookies have a double layer structure, with a harder crust and a softer inner part. In Figure 5 the higher peaks of the curves represent the force applied to penetrate the upper and lower crust [6]. The curves kept this shape during storage, although the magnitude of computed parameters changed. The evolution of texture parameters over time is reported in Figure 6. CO presented starting lower “hardness” and “WOD” values, compared to TU, TX and AR, which do not differ. Samples do not differ in crust thickness. From 15 days and to 30 days, CO cookies had significantly higher hardness values. At 15 days CO samples showed significantly higher values of WOD, in respect to other samples. This can be related to the lower moisture values. Moreover, the higher water activities may reflect a higher degree of sucrose crystallization, which affected texture. Overall, the WOD and Hardness increased during shelf life, reaching their apex at 60 days. From 15 to 30 days, the texture parameters do not show important variations. At intermediate moisture, sucrose recrystallization phenomena and water diffusion to the head space are the main processes involved. As sucrose crystallizes, its structure changes from rubbery to grainy. Sucrose recrystallization frees bound water, with a resulting increase in a_w_ values. This free water is gradually lost due to humidity gradients between headspace, crust and inner part of cookies, leading to an increase in hardness, since water in food acts as a plasticizer, i.e., decreases the viscosity of the material.

### 3.6. Sensory Analyses

#### 3.6.1. Sensory Profile

The sensory attributes evaluated are reported in Appendix A. Results of the sensory profiling are shown as Appendix A at 1, 7, 30 and 60 days of shelf life, respectively.

At T1, as reported, only one attribute related to the visual and tactile aspect (tactile-hardness), and two related to the texture (hardness and adhesiveness), showed significant differences between samples. After 7 days of storage, the samples did not show significant differences for all attributes. The olfactory characteristics, odor and aroma, are comparable to the T1.

Appendix A shows the average values and standard deviations of sensory attributes assessed on amaretti at 30 days of storage. The sensory characteristics tactile-hardness, odor of amaretto and the hardness measured in the mouth, significantly changed. The tactile-hardness reached the intensity value 8 for the TX variety, while hardness measured during chewing, increased, going from 3.5 at 7 days of shelf life, to 5.85 at 30 days.

Appendix A shows data related to amaretti at 60 days of storage. The only attributes that showed significant differences were hardness and friability. The sample with the highest value of both hardness and friability was the one produced with the CO variety.

#### 3.6.2. Consumer Testing

The results of the acceptability test carried out during the storage showed a decreasing trend (Appendix A) in the hedonistic scores, throughout the shelf life. There are no significant differences between samples, at the same time of analysis, as regards the acceptability score relating to the smell/fragrance and taste/flavor, except for T1 and T60. With regards to consistency and global acceptability, the differences between samples at the same time of analysis are significant at T1, T30 and T60. It is possible to observe how the score obtained on overall acceptability decreases with the progress of time. At T1, AR and TX samples have the higher scores. At 30 days, they had the same scores as for the TU. CO samples resulted the least appreciated up to 30 days. Upon reaching T60, AR and TU sample where below the limit of acceptability, TX and CO were at the limit between acceptability and non-acceptability.

## 4. Discussion

The present work was aimed to address the question if almond varieties have a significant effect on the quality features of derived products. Almonds are rich in proteins and lipids, which can affect the texture, flavor and physical chemical characteristics of the final product. The compositional analyses showed no differences in the protein content between varieties, except for TX, that showed the lower value. More marked differences were found in the fat fraction, with TX and CO samples having significantly higher crude fat content and MUFA in the fatty acids profile, compared to the other varieties. Cookies, being intermediate moisture food rich in sucrose, undergo to physical chemical changes during the shelf life. Water loss from the product to the headspace is due to moisture gradients, boosted by increase in a_w_ related to sucrose crystallization. Such changes affect the texture and finally the shelf life and acceptance. Amaretti cookies are characterized by presenting surface breaks, which can potentially represent ways out of the water. This aspect have been investigated by using a specifically developed tool for particles size distribution analyses. The best-fitting procedure was carried out, following the methodology reported in [28], originally conceived for the analysis of gas-cell size distribution in wheat dough. Since there are no differences in the shape parameter, σ*, the distribution of the CKA follows the same shape among samples. On the other hand, μ*, the median of the log-normal distribution, and the geometric mean of the untransformed data, is significantly higher in CO samples. Since no differences were found in the tail of the distributions, the major differences between samples are in the small to medium size CKA. TU and AR samples presented the same μ* value, while TX presents the lower values. CKA could act as preferential ways for moisture exchanges in amaretti cookies, so that differences in the parameter μ* of the CKA, and the total area affected by cracking, could give an account of the differences between samples, regarding water loss and water migration between components, during the shelf life. Color was also affected by the variety used. Color develops during cooking, due to Maillard reactions and caramelization. Maillard reactions takes place in several steps, involving reducing sugars and amino acids, proteins, and/or other nitrogen-containing compounds, when they are heated together, while caramelization refers to complex group of reactions that occur due to direct heating of carbohydrates at higher temperatures, in particular sucrose and reducing sugars [29]. In cookies, color development is strictly related to the browning reaction, which occurs especially on the surface, were a_w_ levels decrease 0.4–0.7 during cooking and temperature surpasses 105–120 °C. Our results showed that the almond variety has an impact on color development, with the major differences found between AR and CO varieties. The differences observed between cookies coming from different varieties can be related to differences in amino acids composition, proteins and sugar content and type [30], and the presence of pigments. The importance of browning development during baking is not only related to sensorial aspects such as color formation but also on flavor generation [31,32], both affecting quality and acceptance. Regarding the texture features, the major differences were found in the intermediate time of analyses. One variety in particular performed worst. From 15 days and to 30 days CO cookies had significantly higher hardness values. At 15 days CO samples showed significantly higher values of WOD, in respect to other samples. At the same time of analysis, the same variety showed the lower moisture level. Descriptive analysis showed differences in few descriptors, particularly tactile, in fresh samples and at 30 days of storage. The sensory characteristics “tactile-hardness”, “odor of amaretto” and “hardness” measured in the mouth, significantly changed. Hardness, both tactile and during chewing, reached the higher intensity in TX. At 60 days, the sample with the highest value of both hardness and friability was the one produced with the CO variety. Amygdalin has a great sensory impact when raw almonds are eaten, while its effect has been attenuated in the processed product, since other ingredients and the use of bitter almonds in the recipe surely had a flattening effect on its olfactory impact.

In an attempt to unscramble the complex interactions between food components, textural features, moisture and a_w_ changes, sensory characteristics and acceptance, a dimensionality reduction approach was used, performing a PCA over data. The first two principal components (PC1 and PC2) explained 72.25% of the variability of the data (Figure 7 and Figure 8).

Variables are distributed in the PCA plan based on their relative contribution to the principal components. Variables taken into account were: water activity (a_w_), moisture content (Moist%), total cracked areas (mm^2^) (CKA), the median of the log-normal distribution of cracking surfaces (μ), CIEL*a*b* color coordinates (L*, a* and b*), instrumental hardness (HRD), crust thickness (THICK), and work of deformation (WOD), sensory scores for appearance (APP), taste (TASTE), flavor (FLV), texture, and overall acceptability (OA), amygdalin content (Amy), crude fat content (Lip), Protein content (Prot), percentage of SFA (SFA%), MUFA % (MUFA %), PUFA (PUFA%). Tactile sensory parameters correlated well with instrumental texture, so we omitted these parameters in the PCA matrix. The PC1 is correlated mostly and positively with Moist % (0.93), APP (0.97), TASTE (0.9748), FLV (0.9862), TXT (0.9463), OA (0.9637), while negatively correlated with HRD (−0.8141), THICK (−0.7456), WOD (−0.9546). Samples with higher sensory scores and “softer” texture plot in the positive part of PC1. On the contrary, samples with lower sensory scores and “harder” texture plot in the negative part of PC1. PC1 (37.96%) can be interpreted in terms of “*acceptability as related to instrumental texture*”. PC2 is correlated positively mostly with a* (0.9136), Amy (0.9167), SFA% (91.98) and PUFA% (986) and negatively with CKA (−0.9299), Lip (−0.9738) and MUFA% (98.39). Samples with higher a*, Amy, SFA% and PUFA%, tend to plot in the positive part of PC2, while samples with higher CKA, Lip and MUFA% should plot in the negative part. This second PC could be interpreted in terms of “*redness, cracking surfaces and composition*”.

“High quality”, intended to maintain high sensorial standards during the shelf life, lasts up to 30 days. TX and TU samples showed close position along the PC1, for all the time of analyses. AR clustered in the positive part of PC1 until day 7, showing sensory acceptance similar to TX and TU. For most of the shelf life, CO samples were characterized by lower sensory scores, higher instrumental hardness, thicker crust and more viscous texture (higher WOD), showing larger Cracked Area, and lower moisture. Samples formed clearly distinguishable clusters along the PC2. Overall, our data showed a marked influence of variety in the physical chemical characteristics of derived products. The correlation matrix showed that Protein content was correlated positively (0.7318) with the parameter *µ*, the median of the cracking surfaces distribution, which is correlated with a third PC (0.9809), although it’s interpretation in relation to physical chemical evolution and sensory acceptance results problematic. Crude fat content was positively correlated with the total area interested by cracking CKA (0.8636). The sensory acceptance, as clearly showed in the loading plot (Figure 7) is inversely correlated with the magnitude of texture features, namely HRD, WOD, THICK, and a_w_, and positively correlated with Moist%. Sensory acceptances were higher at 7 days for all the samples, and similar between samples until day 30, were CO samples showed the lower values. Sixty days was selected as the end of the shelf life for all samples.

## 5. Conclusions

The present study showed that the almond variety affects the quality and shelf life of derived sweets. The analyzes carried out show differences in the evolution of the chemical, physical and sensory characteristics of the analyzed products. The image analysis tool created and illustrated in this work is useful for studying the influence of the fracture surfaces and their correlation with the other measured quantities. The fat content appears to be correlated with the extent of surface cracks, and influences their distribution in dimensional frequency classes. Multivariate analysis showed that there is a high correlation between the acceptability of the product and the intensity of some tactile attributes and moisture content, and also divided the varieties according to the characteristics of color, fracture surfaces and chemical composition. Overall, our results show how the choice of the variety to be used in this type of product is extremely important, directing producers and processors towards those with the best aptitude for transformation, given the same agronomic performances. 

## Figures and Tables

**Figure 1 foods-09-01272-f001:**
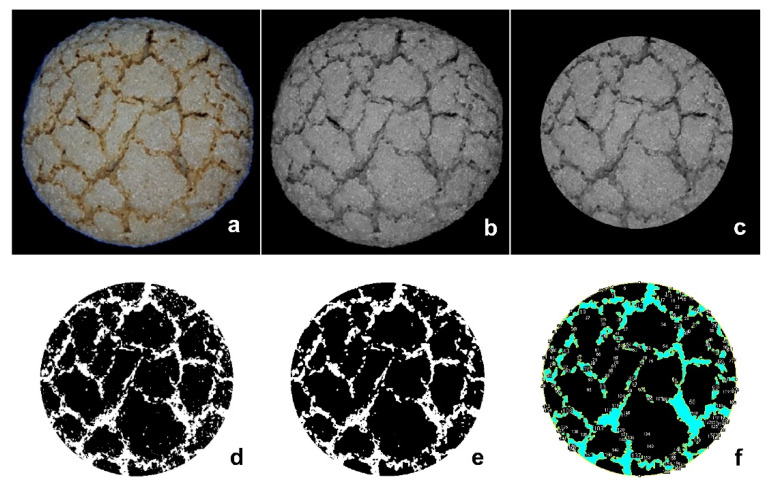
Image processing steps: (**a**) Original RGB image; (**b**), Greyscale converted; (**c**), Selection of the area of interest; (**d**) Thresholding (Otsu); (**e**), Noise reduction; (**f**), Particles analysis.

**Figure 2 foods-09-01272-f002:**
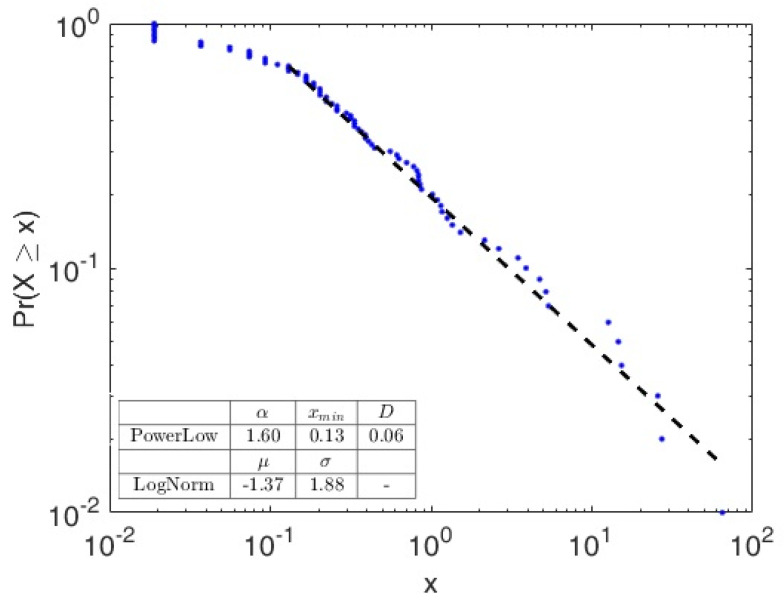
Example of Best fitting analysis results plot of 1 sample data. Complementary Cumulative Density Function (y axis) and areas frequencies (x axis) are in log scale. The dashed line is the Power law distribution tail computed.

**Figure 3 foods-09-01272-f003:**
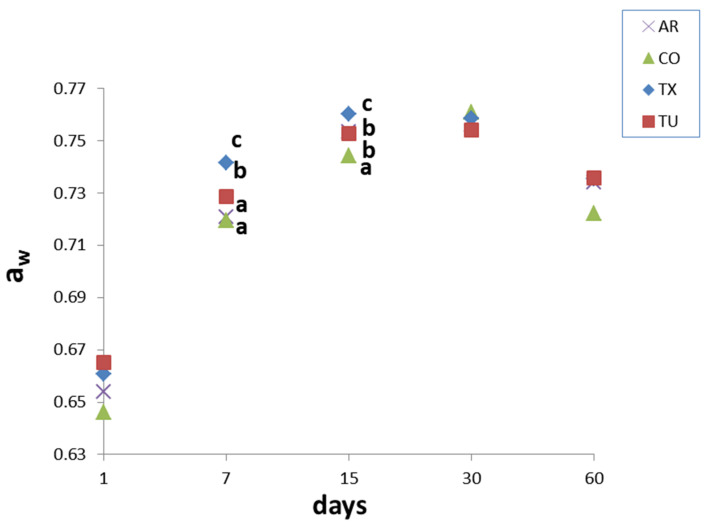
a_w_ values during shelf life, mean values. Values sharing the same superscript letter do no differ significantly (*p* ≤ 0.05), according to Tukey’s HSD post hoc test.

**Figure 4 foods-09-01272-f004:**
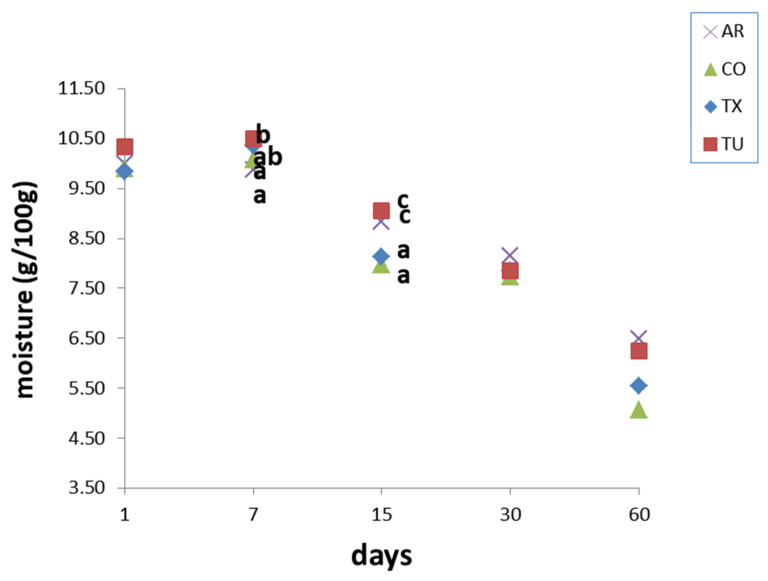
Moisture values during shelf life, mean values. Values sharing the same at the same time letter do no differ significantly (*p* ≤ 0.05), according to Tukey’s HSD post hoc test.

**Figure 5 foods-09-01272-f005:**
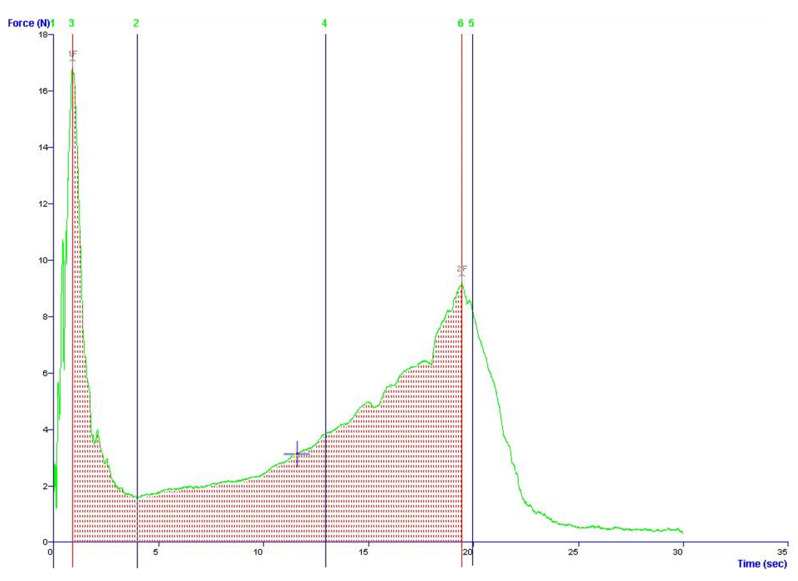
Representative force-deformation curve and estimated parameters. Force at “3”: Hardness (N); Area “3–6”: Work of deformation (N mm); Area “1–3” (sec = mm at 1 mm/sec test speed): Upper crust thickness (mm).

**Figure 6 foods-09-01272-f006:**
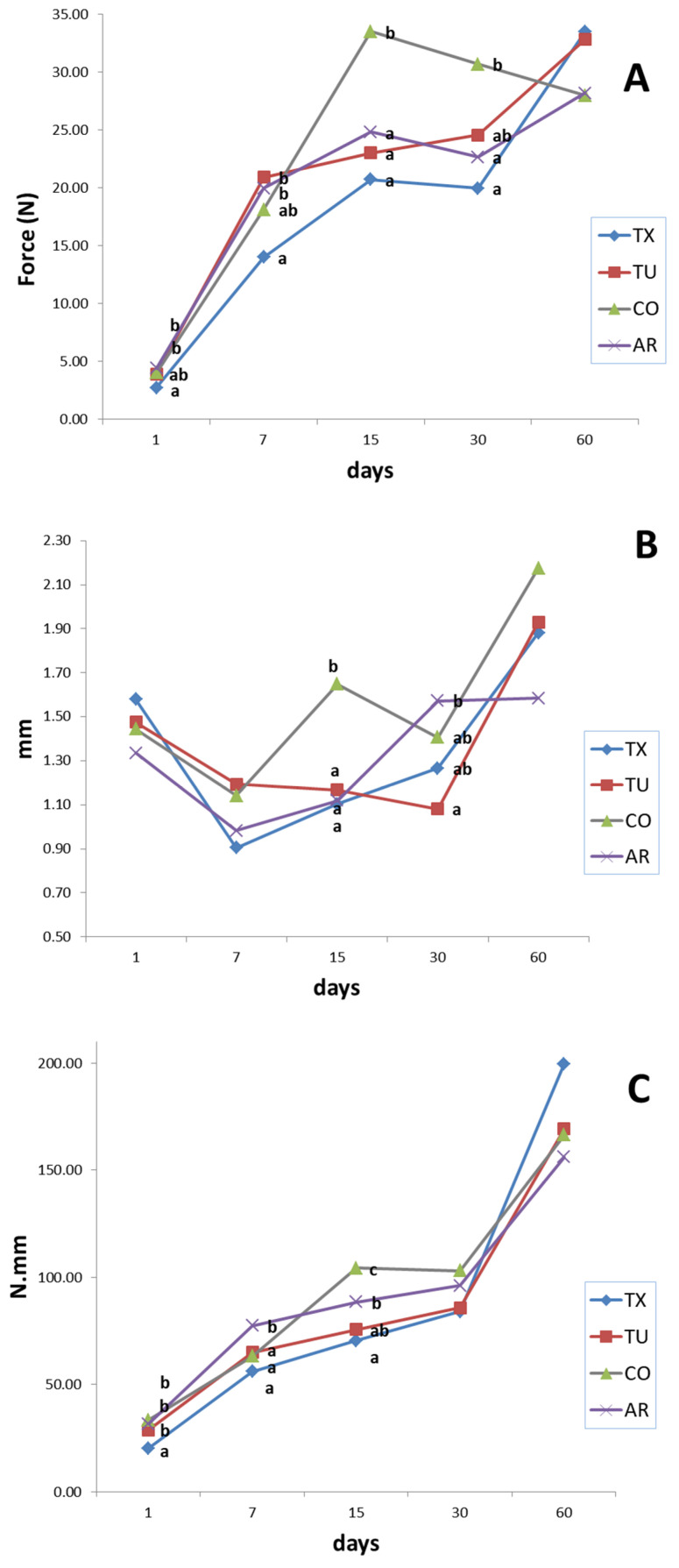
(**A**) Crust Hardness, (**B**) Crust Thickness, (**C**) Work of deformation, Mean values. Values sharing the same letter at the same time do no differ significantly (*p* ≤ 0.05), according to Tukey’s HSD post hoc test. AR; triangle, CO; square, TU; circle, hexagon, TX.

**Figure 7 foods-09-01272-f007:**
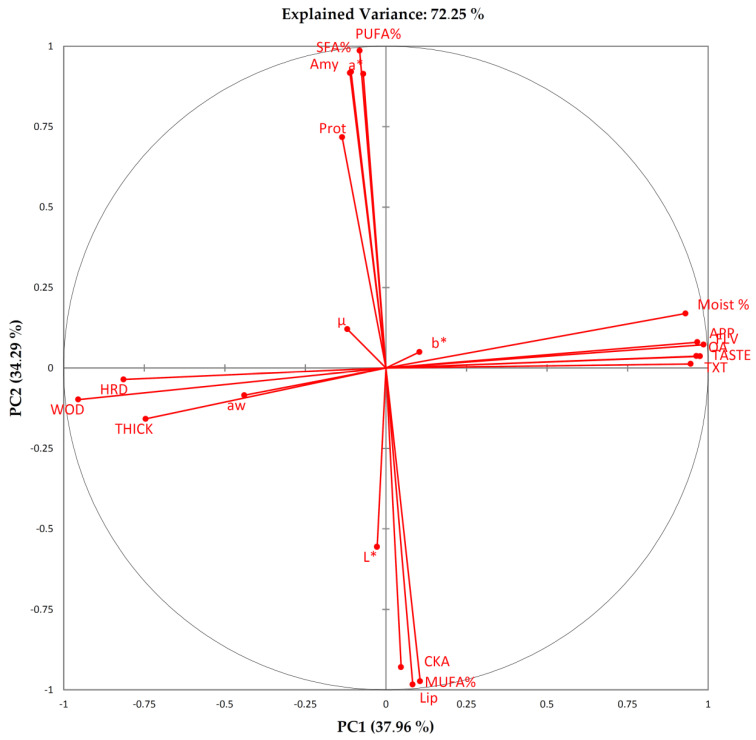
Loading plot. Percent explained variance of the PC are given in brackets.

**Figure 8 foods-09-01272-f008:**
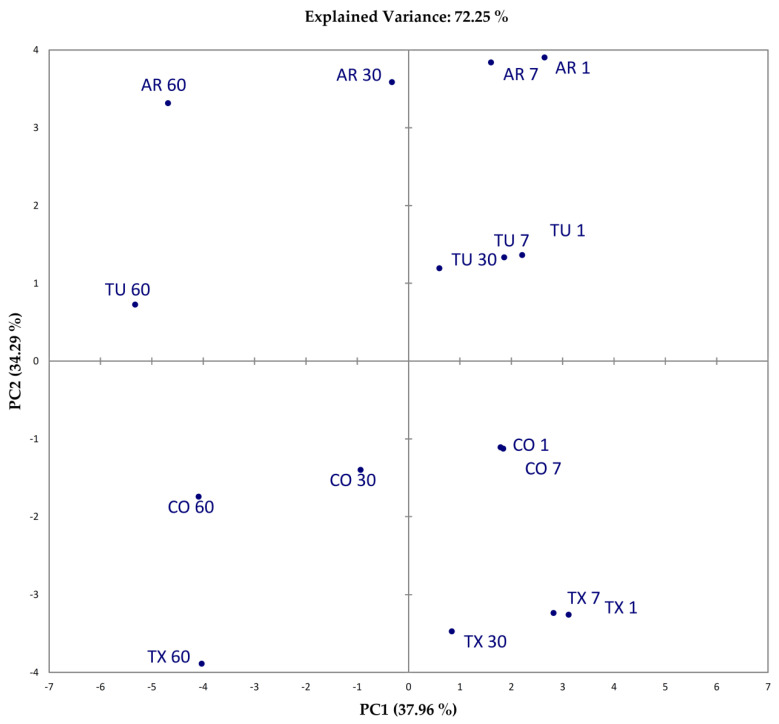
Score plot. Percent explained variance of the PC are given in brackets.

**Table 1 foods-09-01272-t001:** Proximate composition of almond samples.

Sample	Moisture (%)	Proteins % (d.m.b.)	Crude Fat % (d.m.b.)	Amygdalin mg/100 g (d.m.b.)
AR	4.35 ^c^	28.37 ^b^	55.01 ^a^	13.74 ^d^
TU	3.92 ^a^	26.56 ^b^	56.79 ^ab^	6.50 ^b^
CO	4.15 ^b^	28.16 ^b^	57.56 ^bc^	7.34 ^c^
TX	4.92 ^d^	23.72 ^a^	60.14 ^c^	<DL ^a^

Mean values. d.m.b.: dry matter base. DL = Detection limit. Mean values sharing the same superscript letter do no differ significantly (*p* ≤ 0.05), according to Tukey’s HSD post hoc test.

**Table 2 foods-09-01272-t002:** Fatty acids profile of almonds oil.

Samples	AR	TU	CO	TX
Palmitic (C16:0)	7.11 ^d^	6.17 ^c^	5.87 ^b^	5.49 ^a^
Palmitoleic (C16:1)	0.61 ^d^	0.37 ^a^	0.45 ^b^	0.49 ^c^
Stearic (C18:0)	1.42 ^a^	2.4 ^c^	2.31 ^b^	2.35 ^b,c^
Oleic (C18:1)	70.08 ^a^	71.75 ^b^	74.66 ^c^	75.79 ^c^
Linoleic (C18:2)	20.58 ^a^	19.08 ^c^	16.5 ^b^	15.53 ^a^
Linolenic (C18:3)	0.02 ^a^	0.02 ^a^	0.02 ^a^	0.02 ^a^
Others	0.18 ^a^	0.21 ^a^	0.19 ^a^	0.33 ^b^
SFA%	8.54 ^c^	8.57 ^d^	8.18 ^b^	7.84 ^a^
MUFA%	70.69 ^a^	72.12 ^c^	75.11 ^b^	76.28 ^c^
PUFA%	20.6 ^c^	19.1 ^b^	16.52 ^a^	15.54 ^a^
MUFA/PUFA	3.43 ^a^	3.78 ^a^	4.55 ^b^	4.93 ^b^

Data are in g/100 g of oil. Mean values sharing the same superscript letter do no differ significantly (*p* ≤ 0.05), according to Tukey’s HSD post hoc test.

**Table 3 foods-09-01272-t003:** Coefficents arising from the Best fitting procedure for Log normal (all the distribution) and Power law fitting (tail) of data.

	Power Law			Log Normal PDF	
Sample	xmin	α	D	μ*	σ*
		ns		***	ns
AR	0.2428	1.70056	0.0647	0.2464 ^b^	3.3987
TU	0.2514	1.66801	0.0636	0.2610 ^b^	3.2390
CO	0.2896	1.66711	0.0638	0.3120 ^c^	3.1888
TX	0.2219	1.67198	0.0581	0.2124 ^a^	3.1990

Mean values. Values sharing the same superscript letter do no differ significantly (*p* ≤ 0.05), according to Tukey’s HSD post hoc test. *** significant for *p* < 0.05, ns: not significant.

**Table 4 foods-09-01272-t004:** Color analyses results.

CIE Lab Coordinates			AR	TU	CO	TX
L*			63.26	64.78	65.20	64.26
a*			8.25	8.25	7.66	7.60
b*			23.70	23.56	23.08	23.85
	**TX-TU**	**TX-CO**	**TX-AR**	**TU-CO**	**TU-AR**	**CO-AR**
ΔE_00_	0.78	0.88	1.04	0.61	1.26	1.68
ΔE_76_	0.88	1.22 *	1.20 *	0.87	1.53 *	2.12 **
ΔL′	0.52	0.93	1.00	0.42	1.52	1.94
ΔC′	0.00	0.69	0.13	0.69	0.13	0.82
Δh′	1.82	0.77	1.71	1.06	0.11	0.95
ΔH′	0.80	0.34	0.76	0.46	0.05	0.41

* the difference is only noticed by an experienced observer. ** the difference is also noticed by an unexperienced observer.

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
