# Peer review of "Impact of Almond Variety on “Amaretti” Cookies as Assessed through Image Features Modeling, Physical Chemical Measures and Sensory Analyses"

_foods, 2020, doi:10.3390/foods9091272_

Round 1

Reviewer 1 Report

Τhe current manuscript is well written and I suggest that it must be published after some minor revisions. My suggestions are:

1.In paragraph 2.1 the authors should give more precise information about the terroir of cultivation area (climate, soil composition e.g clay, sand, pH, etc). These information are important for understanding the quality of almonds.

2. Supelco EC-1000 capillary column: Please check the column. Is it coded correct? Please provide a chromatogram.

3. Paragraph 2.3: The sum of the recipe ingredients in not equal to 1000. Please define the rest ingredients used. It is very important in order to understand the conlusions.

4. Table 1: Please define the DL value.

5. Table 2: The sum of fatty acid is not equal to 100%. Please correct them or provide the rest fatty acid.

6. pH values must also be given since it is an important parameter to estimate shelf life.

Author Response

Reviewer 1

  1. In paragraph 2.1 the authors should give more precise information about the terroir of cultivation area (climate, soil composition e.g clay, sand, pH, etc). These information are important for understanding the quality of almonds.

Line 71-72, we added the following text: “The soil was sandy-clay (42% sand), pH 7.4,  the average annual precipitations (5-year) were 475.4 mm.”

  1. Supelco EC-1000 capillary column: Please check the column. Is it coded correct? Please provide a chromatogram.

That was an error, thank you for noticing. The column used was a SP-2380 Capillary GC Column (Supelco, Pennsylvania, USA; 60 m x 0.25 mm i.d.; 0.2 µm film thickness). The column name was corrected within the text. Due to restrictions in access to laboratories, due to COVID, we cannot provide a chromatogram at the moment.

  1. Paragraph 2.3: The sum of the recipe ingredients in not equal to 1000. Please define the rest ingredients used. It is very important in order to understand the conclusions.

Apologies. We missed to add “citrus aroma” to the list of ingredients. We added “4 g of citrus aroma” in the text.

  1. Table 1: Please define the DL value.

With the acronym DL we are referring to the “Instrumental Detection Limit” as the lowest concentration (the smallest amount) of an analyte that can be detected, but not quantified, by means of the measuring instrument. Since no clear peak for amygdalin was visible in the chromatogram, in respect to noise, we assumed that the analytic concentration was below the detection limit. We have not  determined such limit, nor will be ease to define this right now, due to Covid 19 restriction.  I hope that this aspect is not prejudicial to the validity of the data presented.

  1. Table 2: The sum of fatty acid is not equal to 100%. Please correct them or provide the rest fatty acid.

Some minor fatty acids generated peaks that were too small to be isolated with certainty from the background noise, so instead of a risky presumptive determination, we decided to put them collectively under the heading "others". Addressing the requests of Reviewer, we added the sum of the minor fatty acids ad “others” in the table (add up to less than 0,3% in all samples).

  1. pH values must also be given since it is an important parameter to estimate shelf life.

It would be possible to measure pH of the dough, prior to cooking, it is not after cooking, or anyway of no use, taking into account that amaretti are characterized by a soft inner part and a hard crust. However,  in this type of products (baked shelf-stable intermediate moisture food) aw values do not support bacterial growth, so pH is not a limiting factor, while moulds could proliferate when aw increases during time. We could address the reviewer request if we had registered such measure, but we did not, taking into account the above considerations on the stability of this type of food.

Reviewer 2 Report

COMMENTS:

The reviewed manuscript is original with sufficient novelty. It has clearly stated the problem being investigated, it means, whether it is possible to extend shelf life of amaretti cookies by the selection of appropriate variety of almonds for its production. Factors limiting the shelf life of amaretti cookies are progressive hardening and sucrose recrystallization. Four varieties were chosen for the study, including two local and two widely cultivated varieties. Almonds were analysed for the content of moisture, protein, fat, amygdalin and fatty acid profile. The changes in qualitative parameters of amaretti cookies, through image data modeling, texture, physical, chemical and sensory analyses, were measured four or five times during two months of storage. A special attention was paid to surface cracks, typical for this type of bakery products, as they can play a role in the shelf life and stability of cookies. In this regards, the Authors developed a workflow for image analysis, which allows to receive a description of the dimensional classes frequencies of the surfaces affected by breakages. This is most valuable part of the manuscript.

The workflow has proven its suitability in the study performed and implies its possible use in other studies of this type or in practise. The results are statistically very well analysed. As the Authors assumed, the variety of almonds had a significant impact on the changes in quality parameters of amaretti cookies during storage. The results, based on consumer testing, allowed the Authors to indicate three varieties of higher acceptability.

The results are presented in four tables and eight figures. Six supplementary tables with the results of sensory evaluation are included.

Critical remarks to the reviewed manuscript refer to the lack of literature references mainly to the methods of chemical analysis used in the study and many language deficiencies.

Only three ingredients were analysed in dehulled almonds, water, protein and fat. Why was the rest of the ingredients not determined, such as minerals, dietary fibre and free sugars. Have these remaining, not determined ingredients some impact on the changes in the quality of amaretto cakes during storage?

 Detailed remarks:

In my opinion, each part of the article needs to be corrected, mainly from the linguistic side (there are many typing errors or incorrect used phrases, for example – almonds origin, lipids content). It would be better, if the tables were easier for the readers to follow. For this, the order of the results presented for the varieties should be the same, for example AR, CO, TX, TU.

Abstract:

Line 17 - Dot missing - Texas).

Line 18 - The evolution……..has been measured…

Line 27 – Cossu, not CO

Ad 1. The introduction is correct and contains enough information about the problem and about the aim of the study.

Ad. 2.1. I think it should be - Almond origin and processing

It would be good to mention at the end of the materials, whether the dehulled almonds were used for both chemical analyses and making cookies.

Ad. 2.2. I think it should be – Almond composition

Ad. 2.2.1. I propose to remove the word – determination

Line 85 - Moisture content of whole almonds – hulled or dehulled? (see Ad 2.1.)

Line 86 – In bracket would be better to write 1 mm   pore   size; milling not crushing

Line 88 – wet gross weight? I suggest to remove it from this sentence.

Line 89 – Is 15 minutes enough to cool samples after drying and before weighing?

Ad. 2.2.2. Crude fat was determined in this study or more precisely ether extract, not lipids. What was the method used?

Ad. 2.2.3. I think it should be – Fatty acid profile

What was the method used?

Ad. 2.2.4. I propose to remove the word – determination. What was the method used?

Ad. 2.2.5. I think it should be – Amygdalin content.

Amygdalin was chromatographically analysed, with HPLC. The reference is missing.

Line 124 – amygdalin

Ad 2.3. Amaretti is written from a small or a capitol letter, and sometimes as ”amaretti”. It should be unified in the entire manuscript.

Ad. 2.3.1. Preparation of Amaretti cookies. Ingredients per 1000 g of dough: sucrose, sweet almonds, egg whites and bitter almonds made up 996 g. Which is the rest to 1000 g. Line 141 – Amaretti were rather baked not cooked. The gram - SI unit symbol is g

Ad. 2.4. It should be: Image analysis

Line 156 – It should be - ImageJ

Ad. 2.5.

Line 168 – It should be – measures

174 – It should be – CIE (8)

Ad. 2.6.

Line 186 – It should be – moisture,   6 ground cookies

Ad.2.7.

Line 192 - Better - Texture measures

Ad. 2.9. The number of replicates in chemical analyses is missing.

In my opinion it would be better if the description of the methods applied in this study were divided into three sections:

2.2. Almond composition

2.3. Amaretti cookie quality characteristics

2.4. Statistical analyses

Then the section 2.3. would follow 2.3.1. Preparation of …..and so on…..up to: 2.3.9. Consumer testing.

TABLES:

Table 1 – Usually, for comparison purposes the content of components in any product is presented on dry matter basis. It would be correct to present the protein and fat content in DM %, as the differences in water content were significant.

Table 2 – Linolenic

FIGURES:

Figure 5 – the title has to be improved together with the figure.

The manuscript is generally written in good language, however there are some errors that have to be corrected. I caught a few of them, but probably not all.

Author Response

Reviewer 2

The reviewed manuscript is original with sufficient novelty. It has clearly stated the problem being investigated, it means, whether it is possible to extend shelf life of amaretti cookies by the selection of appropriate variety of almonds for its production. Factors limiting the shelf life of amaretti cookies are progressive hardening and sucrose recrystallization. Four varieties were chosen for the study, including two local and two widely cultivated varieties. Almonds were analysed for the content of moisture, protein, fat, amygdalin and fatty acid profile. The changes in qualitative parameters of amaretti cookies, through image data modeling, texture, physical, chemical and sensory analyses, were measured four or five times during two months of storage. A special attention was paid to surface cracks, typical for this type of bakery products, as they can play a role in the shelf life and stability of cookies. In this regards, the Authors developed a workflow for image analysis, which allows to receive a description of the dimensional classes frequencies of the surfaces affected by breakages. This is most valuable part of the manuscript.

The workflow has proven its suitability in the study performed and implies its possible use in other studies of this type or in practise. The results are statistically very well analysed. As the Authors assumed, the variety of almonds had a significant impact on the changes in quality parameters of amaretti cookies during storage. The results, based on consumer testing, allowed the Authors to indicate three varieties of higher acceptability.

The results are presented in four tables and eight figures. Six supplementary tables with the results of sensory evaluation are included.

Critical remarks to the reviewed manuscript refer to the lack of literature references mainly to the methods of chemical analysis used in the study and many language deficiencies.

Only three ingredients were analysed in dehulled almonds, water, protein and fat. Why was the rest of the ingredients not determined, such as minerals, dietary fibre and free sugars. Have these remaining, not determined ingredients some impact on the changes in the quality of amaretto cakes during storage?

 Detailed remarks:

In my opinion, each part of the article needs to be corrected, mainly from the linguistic side (there are many typing errors or incorrect used phrases, for example – almonds origin, lipids content). It would be better, if the tables were easier for the readers to follow. For this, the order of the results presented for the varieties should be the same, for example AR, CO, TX, TU.

Abstract:

Line 17 - Dot missing - Texas).

A dot has been added.

Line 18 - The evolution……..has been measured…

Replaced “have” with “has”

Line 27 – Cossu, not CO

Replaced “CO” with “Cossu”

Ad 1. The introduction is correct and contains enough information about the problem and about the aim of the study.

Ad. 2.1. I think it should be - Almond origin and processing

Replaced “Almonds” with “Almond”

It would be good to mention at the end of the materials, whether the dehulled almonds were used for both chemical analyses and making cookies.

Line 82 - We added the following text:  Peeled almond kernels have been used for both making cookies and chemical analyses.

Ad. 2.2. I think it should be – Almond composition

Replaced “Almonds” with “Almond”

Ad. 2.2.1. I propose to remove the word – determination

The word “determination” have been removed

Line 85 - Moisture content of whole almonds – hulled or dehulled? (see Ad 2.1.)

Line 86, We do not know the meaning of “Dehulled”. Moisture was determined on peeled almond kernels, which were used for both making cookies and chemical analyses.

We added the word “peeled” as follows:

….For each variety, 35 whole peeled almonds…

Line 86 – In bracket would be better to write 1 mm   pore   size; milling not crushing

Replaced “1000 microns mesh” with “1 mm pore size”

Line 88 – wet gross weight? I suggest to remove it from this sentence.

The sentence “wet gross weight”  has been removed

Line 89 – Is 15 minutes enough to cool samples after drying and before weighing?

Yes, it was. It took a few minutes to bring the samples back to room temperature.

Ad. 2.2.2. Crude fat was determined in this study or more precisely ether extract, not lipids. What was the method used?

We used the term lipid as “any of a diverse group of organic compounds including fats, oils, hormones, and certain components of membranes that are grouped together because they do not interact appreciably with water” but soluble in organic solvents. As reported, the method used is an automated Soxhlet-type extraction (percolation of lipids from sample via continuous evaporation/condensation of the solvent in a closed apparatus. The extract accumulates in the bottom flask, and weighted after solvent evaporation).

Altough literature extensively report the Soxhlet method as a method for “Lipids recovery” or “Total Lipids” determination. we agree with the reviewer and  changed “lipids” for “ether extract” in the title. We changed “lipids” for “crude fat” within the abstract, text (were pertinent)  and in table 1.

Ad. 2.2.3. I think it should be – Fatty acid profile

Replaced “Fatty acids composition “ with “Fatty acid profile”

What was the method used?

The method was lab developed. The method reported is a routinely used for fatty acids profiling of olive oil samples. After derivatization of fatty acids to the corresponding methyl esters, they are submitted to GC separation and identification.

Ad. 2.2.4. I propose to remove the word – determination. What was the method used?

We deleted the word “determination”.  The method used is reported in the paragraph, is was the BCA-Bicinchoninic acid assay, performed using a commercial assay. The method was performed according to the kit instructions.

Ad. 2.2.5. I think it should be – Amygdalin content.

We deleted  “HPLC analysis of” from the sentence.

Amygdalin was chromatographically analysed, with HPLC. The reference is missing.

The method was lab developed, basing on literature. We found  no “Standard” or “reference” method for amygdalin content extraction and determination in fruit kernels. Extraction could be done in water, ethanol, methanol, both boiling or at room temperature. We opted for room temperature and prolonged extraction time in order to avoid the formation of artifacts (neoamygdalin), which can occur for prolonged extraction at high temperatures, obtaining  an high yield at the same time. Separation conditions were established performing preliminary analyses,  in order to test elution gradients and obtain optimal separation and peak symmetry. Starting from an original mixture of CH3CN 20:80 at a column temperature of 40 ° C, various tests were performed, varying the operating conditions (composition of the mixture and temperature of the chromatographic column) to try to improve the chromatographic yield. Based on the test results, it was decided to set the parameters to perform the HPLC analyzes. Added the following references to the reference list:

8. Arrázola1G.; Sánchez, R.; Dicenta, F.; Grané N. Content of the cyanogenic glucoside amygdalin in almond seeds related to the bitterness genotype. Agron. colomb. 2012, 30, 260-265

9.Sánchez-Pérez, R.; Arrázola, G.; Martín, M.L.; Grané, N.; Dicenta, F. Influence of the pollinizer in the amygdalin content of almonds. Sci. Hortic. 2012, 139, 62-65, doi:10.1016/j.scienta.2012.02.028

Line 124 – amygdalin

Corrected “Amygdalin” to “amygdalin”.

Ad 2.3. Amaretti is written from a small or a capitol letter, and sometimes as ”amaretti”. It should be unified in the entire manuscript.

We changed “Amaretti” to “amaretti” in the whole text, except for opening words of sentences following  a dot, which were kept uppercase.

Ad. 2.3.1. Preparation of Amaretti cookies. Ingredients per 1000 g of dough: sucrose, sweet almonds, egg whites and bitter almonds made up 996 g. Which is the rest to 1000 g. Line 141 – Amaretti were rather baked not cooked. The gram - SI unit symbol is g

Apologies. We missed to add “citrus aroma” to the list of ingredients. We added “4 g of citrus aroma” in the text.

 Ad. 2.4. It should be: Image analysis

We corrected  “analisys” to “analysis”.

Line 156 – It should be - ImageJ

Corrected “Imagej” to “ImageJ”

Ad. 2.5.

Line 168 – It should be – measures

We changed “color” to “Color measures”

174 – It should be – CIE (8)

The correct extended name is CIE L * a * b * 1976 color space (anche CIELAB). Often reported as CIE L*a*b* color space.

Ad. 2.6.

Line 186 – It should be – moisture,   6 ground cookies

We changed “Moisture” to “moisture”

Ad.2.7.

Line 192 - Better - Texture measures

We changed “Instrumental Texture measures” to “Texture measures”.

Ad. 2.9. The number of replicates in chemical analyses is missing.

We added the number of analystical replicates “n=5” within brachets  “……Tukey test (n=5, P≤0.05)…

In my opinion it would be better if the description of the methods applied in this study were divided into three sections:

2.2. Almond composition

2.3. Amaretti cookie quality characteristics

2.4. Statistical analyses

Then the section 2.3. would follow 2.3.1. Preparation of …..and so on…..up to: 2.3.9. Consumer testing.

According to the reviewer request, we have divided the materials and method in 3 sections, 2.2. Almond composition, 2.3. Amaretti cookie quality characteristics, 2.4. Statistical analyses.

TABLES:

Table 1 – Usually, for comparison purposes the content of components in any product is presented on dry matter basis. It would be correct to present the protein and fat content in DM %, as the differences in water content were significant.

We have changed the table and caption according to the reviewer’s request, expressing data in dry matter base. However, ANOVA and post hoc test gave the same results in terms of differences between samples.

Table 2 – Linolenic

We corrected the table 2 changing “Linonelico” to “Linolenic”. Some minor fatty acids generated peaks that were too small to be isolated with certainty from the background noise, so instead of a presumptive determination, we decided to put them collectively under the heading "others". Addressing the requests of Reviewer 1, we added the sum of the minor fatty acids ad “others” in the table, to reach 100%.

FIGURES:

Figure 5 – the title has to be improved together with the figure.

Figure five was remade at a higher resolution. The caption text appeared “corrupt”, we fixed that too.

The manuscript is generally written in good language, however there are some errors that have to be corrected. I caught a few of them, but probably not all.